



# Revisiting parameter sensitivities in the Variable Infiltration Capacity model

Ulises M. Sepúlveda[1], Pablo A. Mendoza[1,2], Naoki Mizukami[3] and Andrew J. Newman[3]

[1]Department of Civil Engineering, Universidad de Chile, Santiago, Chile
[2]Advanced Mining Technology Center, Universidad de Chile, Santiago, Chile
[3]National Center for Atmospheric Research, Boulder, CO, USA

*Correspondence to*: Pablo A. Mendoza (pamendoz@uchile.cl)

**Abstract.** Despite the Variable Infiltration Capacity (VIC) model being used for decades in the hydrology community, there are still model parameters whose sensitivities remain unknown. Additionally, understanding the factors that control spatial

variations in parameter sensitivities is crucial given the increasing interest to obtain spatially coherent parameter fields over large domains. In this study, we investigate the sensitivities of 43 soil, vegetation and snow parameters in the VIC model for 101 catchments spanning the diverse hydroclimates of continental Chile. We implement a hybrid local-global sensitivity analysis approach, using eight model evaluation metrics to quantify sensitivities, with four of them formulated from runoff time series; two characterizing snow processes, and the remaining two based on evaporation processes. Our results confirm an

over-parameterization for the processes analysed here, with only 12 (i.e., 28%) parameters found as sensitive, distributed among soil (7), vegetation (2) and snow (3) model components. Correlation analyses show that climate variables – in particular, mean annual precipitation and aridity index – are the main controls on parameter sensitivities. Additionally, our results highlight the influence of the leaf area index on simulated hydrologic processes – regardless on the dominant climate types – and the relevance of hard-coded snow parameters. Based on correlation results and the interpretation of spatial sensitivity

patterns, we provide guidance on the most relevant parameters for model calibration according to the target processes and the prevailing climate type. Overall, the results presented here contribute to improved understanding of model behaviour across watersheds with diverse physical characteristics that encompass a wide hydroclimatic gradient from hyper-arid to humid systems.

## 1.  Introduction

Over the past four decades, the increasing demand for more realistic spatial representations of water storages and fluxes across large domains has motivated the development of more complex physics-based models (e.g., Niu et al., 2011; Clark et al., 2015; Lawrence et al., 2019). The progress in this field has been partly facilitated by new observational datasets (e.g., McCabe et al., 2017; Berg et al., 2018) and advances in computing (see discussion on tradeoffs in Clark et al., 2017), enabling hydrological characterizations at national (e.g., Tian et al., 2017; Zink et al., 2017), continental (e.g., Xia et al., 2012; Abbaspour et al.,

2015) and global (e.g., Schmied et al., 2014; Arheimer et al., 2020) domains.





Although spatially constant parameters can be used for large domain applications, improving model realism requires the specification of parameter values that reflect spatial heterogeneities in landscape properties. Because increasing model complexity is often associated with a larger number of parameters, many models rely on lookup tables to assign soil thermal and hydraulic parameters, and vegetation optical and physiological parameters for each modelling unit (e.g., Mitchell et al.,
2004; Yang et al., 2011). However, parameter uncertainties may be considerable (Rosero et al., 2010; Hou et al., 2012), and such problem is exacerbated by the existence of parameters that, despite being "adjustable" (e.g., runoff generation parameters), remain fixed and hard-coded (Mendoza et al., 2015a; Cuntz et al., 2016).

To address overparameterization problems that are typical in environmental models, sensitivity analysis has become a key tool that provides information on which parameter values are the most influential for the dynamics of specific model responses
(Razavi and Gupta, 2015). The outcomes of sensitivity analysis not only help to improve understanding of model functioning, but also to inform decisions regarding parameter estimation problems. The literature provides many examples of sensitivity analysis studies with process-based hydrological models, including the Biosphere-Atmosphere Transfer Scheme, BATS (Bastidas et al., 1999); TOPKAPI (Foglia et al., 2009); PRMS (Mendoza et al., 2015b), the Noah land surface model (Bastidas et al., 2006; Rosero et al., 2010); Noah-MP (Mendoza et al., 2015a; Cuntz et al., 2016), the Simple Biosphere model, SiB3
(Prihodko et al., 2008; Rosolem et al., 2012); the MESH modelling system (Razavi and Gupta, 2016), the Community Land Model, CLM (Göhler et al., 2013; Massoud et al., 2019) and the Variable Infiltration Capacity (VIC) model (e.g., Demaria et al., 2007; Melsen et al., 2016), which is one of most popular modelling platforms in the hydrology community (Addor and Melsen, 2019).

The VIC model (Liang et al., 1994; Hamman et al., 2018) has been used for myriad applications all over the world, including
snow modeling (Andreadis et al., 2009; Chen et al., 2014), streamflow forecasting (Wood et al., 2005; DeChant and Moradkhani, 2014), water balance studies (Mizukami et al., 2016; Vásquez et al., 2021), extreme event characterization (Melsen et al., 2019); land use change impacts (Chawla and Mujumdar, 2015) and climate change impact assessments (e.g., Vano and Lettenmaier, 2014; Chegwidden et al., 2019). Despite the large number of parameters contained in VIC – either 'free' (e.g., the infiltration shape parameter '*INFILT*', the exponent in baseflow curve) or 'observable' (e.g., leaf area index) –
, many studies have relied on the calibration of only two or three soil-related parameters (Huang and Liang, 2006; Chawla and Mujumdar, 2015). Conversely, other authors have advocated for characterizing parameter sensitivities using different approaches, sensitivity metrics, and including different parameters. For example, Liang & Guo (2003) assessed the sensitivity of annual runoff, annual evapotranspiration (ET), annual mean soil moisture, and annual mean sensible heat flux to variations in five soil and vegetation parameters at three experimental locations (i.e., point scale), finding that sensitivities varied with
climatic and physiographic site characteristics. Demaria et al. (2007) examined sensitivities in simulated catchment-scale runoff responses using lumped VIC configurations, a Monte Carlo method, and five objective functions computed for four basins with varying hydroclimates. They concluded that (i) three (out of ten) soil parameters dominated the simulated runoff response; (ii) the *INFILT* parameter and the drainage parameter (Expt$_i$) depended strongly on local hydroclimatology, and (iii) that the baseflow model formulation is overparameterized.



Subsequent studies aiming at calibrating the VIC model to simulate observed catchment-scale responses have revisited its parametric sensitivity. Mendoza et al. (2015b) applied the distributed evaluation of local sensitivity analysis method (DELSA, Rakovec et al., 2014) to find the parameters that provided the largest sensitivities in root mean squared errors (RMSE) between simulated and observed streamflow; they showed that 9 (out of 34) parameters provided the largest sensitivities in three subcatchments from the Upper Colorado River basin. Melsen et al. (2016) also used the DELSA method to find influential

parameters in three catchments located in Switzerland, identifying four (out of 28) very sensitive parameters for three calibration metrics. Wi et al. (2017) applied the method of Morris (1991) to quantify parameter sensitivities on the Nash-Sutcliffe efficiency (NSE; Nash and Sutcliffe, 1970) with daily flows, finding 6 soil parameters and two temperature threshold parameters as the most influential (out of 15). Gou et al. (2020) characterized the sensitivities provided by 13 soil parameters across 14 catchments in China, finding that *INFILT*, *Depth1* and *Depth2* dominated streamflow responses. Lilhare et al. (2020)

applied the Variogram Analysis of Response Surfaces (VARS) method (Razavi and Gupta, 2016) to examine the sensitivities of three streamflow performance metrics to variations in six soil parameters across 10 catchments in Canada, finding that *INFILT* and *Depth2* parameters dominated streamflow responses. Yeste et al. (2020) quantified relative sensitivities provided by five soil parameters to water balance components across 31 basins in the Iberian Peninsula, concluding that *INFILT* and *Depth2* control runoff components and evapotranspiration (ET). Finally, Melsen and Guse (2021) characterized VIC parameter

sensitivities for a historical (1985-2008) and future (2070-2093) period in 605 catchments of the conterminous United States finding that, in the historical period, *Rmin*, *Depth2* and *Expt2* controlled average streamflow, while *Ds*, *DsMAX*, and many more parameters influenced streamflow timing. Melsen and Guse (2021) also projected increased (decreased) sensitivities to *Depth2* (snow parameters) for the future period when examining average streamflow, and increased (decreased) future sensitivities to deep soil (snow) parameters when looking at discharge timing.

Table 1 summarizes the main characteristics of parameter sensitivity studies with VIC. Note that we have excluded a recent study conducted by Sheikholeslami et al. (2021), who quantified parameter sensitivities in modified version of VIC – specifically, using a slow linear reservoir (Gharari et al., 2019) model instead of the traditional ARNO formulation. Most the studies listed in Table 1 focused on streamflow responses, attributing the largest sensitivities to a few soil parameters (Demaria et al., 2007; Gou et al., 2020; Lilhare et al., 2020). Only two studies – also characterizing streamflow responses – have included

a large number of soil, vegetation and snow related parameters (Melsen et al., 2016; Mendoza et al., 2015b). Additionally, only two studies (Chaney et al., 2015; Bennett et al., 2018) aimed to characterize sensitivities across model grid cells. Chaney et al. (2015) quantified the effects of 9 parameters on annual flow biases, runoff seasonality and daily flow extremes at 1° resolution grid cells across the globe. Bennett et al. (2018) examined sensitivities of projected changes in water balance components to variations in 46 VIC parameters, across a suite of ~7-km grid cells in the Colorado River basin. Sensitivity

analyses at higher resolutions is particularly relevant for the hydrology community, considering recent developments in meteorological datasets (e.g., Tang et al., 2021), global, gridded runoff datasets (e.g., Do et al., 2018; Ghiggi et al., 2019) and the increasing interest to improve the calibration density – i.e., use high resolution data in the calibration process – in distributed hydrology and land surface models (Yang et al., 2019).



In this paper, we quantify VIC parameter sensitivities across 5,574 grid cells (0.05°×0.05°) covering 101 catchments located
in continental Chile, including a suite of 43 standard and hard-coded parameters, and a set of metrics that span different runoff,
ET and snow processes. The results presented here contribute to improved understanding of model behaviour across
watersheds with diverse physical characteristics, spanning from hyper-arid to humid hydroclimates. With this, we seek to
answer the following research questions:

1.  Are there other vegetation and snow parameters, either standard or hard-coded, affecting simulated runoff responses
in VIC?
2.  What are the effects of standard and hard-coded parameters on other simulated processes?
3.  How do parameter sensitivities relate with local climatic and physiographic characteristics?

## 2. Study domain and data

In this work, we select 101 catchments with near-natural hydrological regimes from the CAMELS-CL data set (Alvarez-
Garreton et al., 2018). The selected basins span a total area of 139,350 km$^2$ – i.e., 19% of the territory of continental Chile –,
and meet the following criteria: (i) a maximum threshold value of 5% for the relationship between the annual volume of water
assigned as permanent consumptive rights and the average annual flow (Table 3 in Alvarez-Garreton et al., 2018), and (ii)
absence of large reservoirs within each catchment. The location, hydroclimatic and land cover characteristics across the domain
are shown in Figure 1, and the descriptors used to characterize the grid cells are listed in Table 2. These catchments cover a
wide range of physiographic attributes, with drainage areas spanning 100-7,500 km$^2$, mean elevations ranging 119 - 4824 m
above sea level, mean slopes varying from 52 to 306 m km$^{-1}$, and markedly different land cover types, ranging from completely
covered by native forest or grassland to fully covered by impermeable land. Moreover, the selected basins represent the
diversity in hydroclimatological conditions across the country. For example, the hydrology of catchment 2101001 (Rio Loa
before Lequena dam, Figure 2a) is influenced by arid conditions between March and November, and Altiplanic winter events
triggering runoff increases between December and March; towards the south, there is a transition from arid to semi-arid
conditions (see progression in Figure 2b-c), with precipitation events occurring mostly during fall and winter (especially May-
August), favoring the accumulation of snow in the headwaters of Andean catchments, and thus snowmelt-driven regimes.
Catchments 7115001 (Palos River at junction with Colorado River; Figure 2d) and 8317001 (Biobio River at Rucalhue; Figure
2e) reflect the transition towards mixed regimes, with larger contributions of winter rainfall events to runoff. Finally, catchment
9129002 (Cautin River at Cajon; Figure 2f) has a rainfall-dominated hydrological regime, with the largest runoff volumes
during the winter season (i.e., June-September).

In this study, meteorological forcing data is obtained from various sources. Time series of daily precipitation and maximum,
average, and minimum daily temperature are obtained from the CR2MET meteorological dataset, introduced in DGA (2017),
which provides data for continental Chile at a horizontal resolution of 0.05°×0.05° (~5 km) for the period 1979-2016. The
precipitation product builds upon a statistical post-processing technique that uses topographic descriptors and simulated





meteorological variables from ERA-Interim (Dee et al., 2011) and ERA5 (C3S and Copernicus Climate Change Service (C3S), 2017) as predictors, and daily precipitation records as the predictand. A similar approach is used to generate time series of daily maximum and minimum temperatures, including additional predictors from MODIS land surface products. Daily precipitation and temperature variables are disaggregated into 3-hourly time steps using the sub-daily distribution provided by
ERA-Interim. Finally, relative humidity and wind speed were obtained from a blend between ERA-Interim and ERA5, which was subsequently rescaled at the CR2MET horizontal grid through spatial interpolation. It is important to note that this product combination was created because ERA5 was not available during the entire study period (1985-2015) at the time of data acquisition (early 2018, where only 2010-2016 data was available). However, the updated reanalysis information, despite the short time coverage, was included due to various developmental improvements.

**3. Methods**

**3.1 Hydrological model**

The Variable Infiltration Capacity (Liang et al., 1994, 1996) model (version 4.1.2.g) is a semi-distributed, physically-based hydrological model that simulates snow accumulation and melt, evapotranspiration (ET), canopy interception, surface runoff, baseflow, and other hydrological processes at daily or sub-daily time steps. In VIC, the domain of interest can be spatially
discretized into grid cells. Sub-grid land-use type variability is accounted for by providing vegetation tiles and the fractional areas, for which water and energy balance equations are solved separately; then, model states and fluxes are spatially averaged to provide results at the pixel scale. In VIC, each grid cell can have up to three soil layers: the two top layers represent the interaction between moisture and vegetation, and the bottom soil layer is used to simulate baseflow processes. VIC does not incorporate an aquifer at the bottom of the soil column, nor lateral exchange of fluxes between grid cells. Finally, snowpack
dynamics are simulated by a two-layer mass and energy balance model (Cherkauer et al., 2003; Andreadis et al., 2009), where the surface layer solves the energy exchange between the snowpack and the atmosphere, and the lower layer stores the excess snow mass from the upper surface layer.

**3.2 Parameters considered for sensitivity analysis**

We considered a suite of 43 parameters (Table 3) to incorporate most of soil, vegetation, and snow processes simulated by
VIC. It should be noted that three of the snow parameters are not exposed to model users (*NEW_ALB*, *ALB_AA* and *ALB_THA*), although the associated relationships and default values were proposed decades ago (USACE, 1956). For those parameters with monthly variations, we examined sensitivities using regularization "superparameters" (Tonkin and Doherty, 2005), also called multipliers (Pokhrel and Gupta, 2010), which are uniformly applied over all monthly values. Hence, multipliers are used for the leaf area index (*LAI*); vegetation albedo (*ALB*); vegetation roughness length (*ROU*) and vegetation displacement (*DIS*).
Despite some of these parameters are considered observable, a non-negligible degree of uncertainty may be involved in their determination; an example is the *LAI* parameter (Tian et al., 2002; Fang et al., 2012, 2013), whose implementation in many





hydrology and land surface models is simplified through static monthly values. Assumptions like this motivate us to explore the sensitivity of this type of parameters, which may have the potential to be included in the calibration process.

Despite the aim to include the largest possible number of parameters, some of them were discarded for different reasons. For example, a few soil parameters (e.g., soil bubbling pressure) are not active unless the frozen soil algorithm is turned on. We also excluded the parameter *trunk_ratio* – i.e., the ratio of total tree height that is trunk (no branches) – because it is activated only in those grid cells with forest (i.e., vegetation class with overstory, spanning 22% of our study domain) as land cover type. Finally, we found five mutually related soil parameters that do not allow independent variations: soil bulk density (*bd*); soil particle density (*sd*); fractional soil moisture content at the critical point ($\theta_{cr}$), fractional soil moisture content at the wilting point ($\theta_{wp}$) and residual soil moisture ($\theta_r$). These parameters are related following:

$$\theta_{cr} \geq \theta_{wp} \geq \frac{\theta_r}{\left(1 - \frac{bd}{sd}\right)},\qquad(1)$$

From these five parameters, we only include $\theta_r$ and *bd* because (1) perturbing $\theta_r$ and *bd* values did not affect numerical solutions, and (2) Bennett et al. (2018) showed that $\theta_{cr}$, $\theta_{wp}$, *bd* and *sd* did not have substantial effects on model simulations. Finally, those parameters that showed little or no sensitivity in the initial phases of the study were purposely discarded.

### 3.3 Sensitivity analysis approach

We used the Distributed Evaluation of Local Sensitivity Analysis (DELSA; Rakovec et al., 2014) method, which is a derivative-based, hybrid local-global approach. DELSA combines elements from the method of Morris (Morris, 1991), the Sobol' method (Sobol', 2001) and regional sensitivity analysis (Hornberger and Spear, 1981), and provides robust results with a fewer number of model simulations compared to variance-based global methods such as Sobol'. Although our implementation only examines first-order sensitivities (as in Mendoza et al., 2015b; Zegers et al., 2020; among other studies) DELSA has unexplored potential to quantify parameter interactions, which could be achieved by including additional terms in the local total variance (Sobol' and Kucherenko, 2010).

Consider a transformation $f$ and a vector $\theta$ with k parameters, which provides a metric $\Psi$ describing model output:

$$\Psi = f(\theta), \quad f: R^k \to R,\qquad(2)$$

Given a sample point $\theta^*$ in the parameter space, the gradient for metric $\Psi$ and parameter $\theta_j$ around this point – i.e., $\frac{\partial \Psi}{\partial \theta_j}|_{\theta^*}$ – is considered a measure of local sensitivity. In this work, we follow Rakovec et al. (2014) and compute such gradient using a forward, finite difference approach with 1% change in the parameter value:

$$\frac{\partial \Psi}{\partial \theta_j}\Big|_{\theta^*} = \frac{\Psi(\theta_j^* + 0.01\theta_j^*) - \Psi(\theta_j^*)}{0.01\theta_j^*}\qquad(3)$$



In equation (3), $\Psi(\theta^*)$ is a signature measure of hydrologic behavior, formulated by contrasting model output at the point $\theta^*$ with that obtained from a reference parameter set $\theta^{ref}$ in the grid cell of interest (see section 3.3.2 for details). The first-order sensitivity measure for the $j^{th}$ parameter is calculated at each sample point as:

$$S_j^L = \frac{\left(\frac{\partial \Psi}{\partial \theta_j}\big|_{\theta^*}\right)^2 s_j^2}{V_L(\theta^*)}, \tag{4}$$

Where $s_j^2$ is the a priori parameter variance of the $j^{th}$ parameter, and $V_L(\theta^*)$ is the linearized local variance:

$$V_L(\theta^*) = \sum_{j=1}^k \left(\frac{\partial \Psi}{\partial \theta_j}\big|_{\theta^*}\right)^2 s_j^2, \tag{5}$$

Finally, $s_j^2$ is obtained from the variance of a uniform distribution (Rakovec et al., 2014), which is $\frac{1}{12}\left(\theta_{j,max} - \theta_{j,min}\right)^2$.

The first-order sensitivity indices vary between 0 and 1, and the sum of first-order sensitivities from all parameters at each sampling point is equal to 1. Local sensitivities can be examined through their cumulative frequency distribution across the parameter space, or by computing a specific statistical property. Here, we quantify the relative contribution of a specific parameter using the area above the curve of the full frequency distribution:

$$IS_j^L = 1 - \int_0^1 F\left(S_j^L\right) dS_j^L \tag{6}$$

### 3.4 Performance metrics

We use eight model evaluation metrics to quantify the sensitivity of simulated hydrological processes to variations in model parameters. The notation, brief description, mathematical formulation, and physical process associated with each metric are detailed in Table 4. These metrics are computed by contrasting model output from sampling points produced for DELSA, with a reference, national scale dataset with simulated states and fluxes obtained from the National Water Balance database (DGA, 2018, 2019, 2020) for the historical period 1985-2015. Such dataset was developed by running the VIC model at the same grid discretization employed here (i.e., 0.05°x0.05°), using a combination of CR2MET version 2.0, ERA-Interim and ERA5 output as meteorological forcings. The spatially distributed parameter fields for our reference simulation were developed via parameter regionalization, based on the similarity between possible donor catchments – whose parameters were calibrated individually (Vásquez et al., 2021) – and each grid cell across the domain, following Beck et al. (2016). The reader is referred to Vásquez et al., (2021) and DGA (2018, 2019, 2020; in spanish) for more details on individual model calibration and parameter regionalization procedures used to generate the reference simulation.

Four evaluation metrics are formulated from runoff time series. The first objective function is the root-mean-square error (RMSE), which is a standard metric that emphasizes high flows. The second metric selected is the transformed-root-mean-square error (TRMSE), for which the simulated and observed runoff time series are transformed using a Box-Cox transformation to emphasize low flows (Misirli et al., 2003). The third objective function is the flow duration curve (FDC)



midsegment slope difference (FMS), which represents the variability, or flashiness, of the flow magnitudes, so it measures how well a model captures the distribution of the mid-level flows. A steep slope of the FDC indicates flashiness of the streamflow response, whereas a flatter curve indicates a relatively damped response and a higher storage (Yadav et al., 2007;

Casper et al., 2012). The fourth evaluation metric is the runoff ratio difference (RR), considered as a measure of the general water balance and, therefore, as a signature of the evapotranspiration model component (Mendoza et al., 2015b).

We use two metrics to characterize snow cover processes: the difference in long-term simulated peak SWE, and the difference in snow cover duration, quantified by the number of days with snow on the ground (Mizukami et al., 2014). Finally, we include two metrics based on evaporation fluxes. The sublimation difference (SUBL) emphasizes the net sublimation from the

snowpack surface, and the plant transpiration difference (TRANSP).

### 3.5  Experimental setup

We apply the DELSA method in 5,574 grid cells across continental Chile, which are contained within the 101 catchments described in section 2. In each grid cell, hydrologic model simulations are conducted at 3-hourly time steps for a 12-year period (April/1999 – March/2011), with the first two years used to initialize model states (i.e., spin-up period). The model is run in

full energy balance mode, which means that both energy and water balances are solved, and 3-hourly outputs are aggregated to daily time steps for subsequent analyses.

In this study, we use the Latin hypercube sampling (LHS) method to obtain 200 sample points across the parameter space, for which first-order sensitivity indices are computed. LHS is an efficient simulation technique, especially suitable for statistical and sensitivity calculations (Vořechovský, 2015). To stratify the parameter space, we sample uniformly in a 43-dimensional

hypercube, and map onto the parameter space using the inverse cumulative distribution function of each parameter's prior. Since DELSA is used here, all parameter distribution functions are assumed to be uniform. The computational cost of applying DELSA at each cell is $N_l$ ($k$ +1) = 8,800 model runs, where $N_l$ is the number of sample points (200) and $k$ is the number of parameters (43), and the total number of models runs required for this study is 49,051,200.

In this paper, a parameter is considered redundant or insensitive when the median value of the integrated first-order sensitivity

index (i.e., median $IS_j^L$) across all grid cells is smaller than 0.05 for at least seven of the eight evaluation metrics listed in Table 4. Parameter sensitivity results are also examined per metric (section 3.4) and grid cell climate type based on the aridity index (UNEP, 1997; Verbist et al., 2010; Table 5).

Finally, we use the Spearman rank correlation coefficient, $r_s$, to measure the degree of association between parameter sensitivities and physiographic/hydroclimatic characteristics listed in Table 2. These attributes were chosen due to their ability

to improve the prediction of hydrological signatures (Addor et al. 2018) and because they are relatively easy to obtain.



## 4. Results and discussion

### 4.1 Intra and inter-basin variability in parameter sensitivities

Figure 3 shows the cumulative distribution functions of sensitivity indices $S_j^L$ for combinations of evaluation metric and parameter. Each panel located in the first six rows includes the CDFs of all the grid cells contained in a specific basin, and the

last row in Figure 3 comprises the CDFs of all grid cells across six hydroclimatically different catchments – also displayed in Figures 1 and 2. The results reveal high parametric sensitivities for RMSE-*INFILT* in basins located in northern Chile (arid regime) and lower sensitivities along central-southern catchments, while the opposite behavior is observed for TRMSE – *DsMAX*, FMS – *DsMAX* and RR – *LAI* (i.e., increasing sensitivities towards the south). Such dependence between hydroclimatic basin characteristics and parameter sensitivities was also reported by Demaria et al. (2007). Gou et al. (2020)

also found that sensitivities were strongly related to environmental characteristics, including climate, vegetation, soil and topographic features. Figure 3 also enables the comparison between intra-catchment (top six rows) and inter-catchment (bottom row) variability in parameter sensitivities. For the sample of basins included here, one can note that inter-basin variability in sensitivities is larger than intra-basin variability in runoff-related metrics. Nevertheless, for some combinations of metric and parameter intra-catchment variability is comparable to inter-basin variations in parametric sensitivities. For example, the

spread in the CDFs displayed for SUBL-*z0_SNOW* at basin 8317001 (Biobio River at Rucalhue) – characterized by a wet hydroclimate – is comparable to the spread arising from all basins (see same column, last row).

To further illustrate intra-basin differences in parameter sensitivities, Figure 4 and Figure 5 show the spatial distribution of the $IS_j^L$ indices for the leaf area index (*LAI*) and the snow albedo parameter *ALB_THA*, respectively, over a cluster of sub-humid and humid basins located in southern Chile. For the *LAI* parameter (Figure 4), a west-east gradient in $IS_j^L$ is observed for RMSE

(high flows), TRMSE (low flows) and FMS (flashiness of runoff), with increasing sensitivity to *LAI* variations towards the coast, while an inverse pattern is observed for the same metrics and *ALB_THA* (i.e., larger sensitivities towards the Andes, Figure 5). For PeakSWE, SUBL, and – to a smaller degree – SnowLength, *LAI* yields larger sensitivities in vegetated areas, where snow accumulates during winter (Figure 4), matching those locations where forest is the dominant land cover type, which is also the only vegetation class with overstory (e.g. trees). Notably, very large variations in PeakSWE sensitivities to

*LAI* are observed over relatively short distances due to differences among grid cells in the fraction of land cover defined as forest. Such dependence among SWE sensitivities, *LAI* and canopy fractions was also reported by Bennett et al. (2018). Figure 4 also shows that LAI does not yield a clear sensitivity pattern in RR and TRANSP throughout this subdomain, although $IS_j^L$ values are higher for RR. For this metric, there are spatial singularities where the sensitivity is minimal or null since, in these areas, the fraction of ground cover defined as bare soil increases considerably, reaching up to 100% of bare soil (*LAI* ~ 0) in

some grid cells.

The results presented in Figure 5 reinforce the idea that hard-coded parameters should be exposed to users (Mendoza et al., 2015a; Cuntz et al., 2016). In particular, Figure 5 shows the large effects of *ALB_THA* variations on SnowLength (with a very





pronounced east-west gradient) and, to a smaller degree, on PeakSWE and SUBL. *ALB_THA* also affects runoff-based metrics along the Andes, especially on simulated high (RMSE) and low (TRMSE) flows.

## 4.2 Identification of most sensitive parameters

Figure 6 displays box plots comprising $IS_j^L$ results from all grid cells in the study domain, for each parameter and evaluation metric (displayed in different panels). The results show that 72% of the parameters analysed (i.e., 31) yield little sensitivities for the metrics examined here. Conversely, a suite of 12 sensitive parameters are associated to soil (*INFILT*, *Ds*, *DsMAX*, *Ws*, *Expt2*, *Depth2*, *Depth3*), snow (*NEW_ALB*, *ALB_THA*, and *ALB_AA*), and vegetation (*Rmin* and *LAI*) processes. Figure 7 shows the spatial variability of $IS_j^L$ for the 12 parameters identified as the most sensitive across the 101 basins of continental Chile.

For the case of high flows (RMSE), low flows (TRMSE), and flashiness of runoff (FMS), the parameters identified as sensitive are *INFILT*, *Ds*, *DsMAX*, *Ws*, *Expt2*, *Depth2*, and *Depth3* (see top three panels in Figure 6). The parameter *INFILT* controls the shape of the variable infiltration capacity curve (Zhao et al., 1980; Wood et al., 1992), and thus the partitioning of rainfall or snowmelt into infiltration and surface runoff. A higher *INFILT* value yields less infiltration and higher surface runoff. The RMSE and TRMSE metrics are particularly sensitive to *INFILT*, indicating a key role in the generation and timing of high and low flows. This parameter has been identified as sensitive in all the studies listed in Table 1. *DsMAX* is the maximum velocity of baseflow, while *Ds* and *Ws* are the fraction of *DsMAX* and the fraction of the maximum soil moisture content in the third layer, respectively, where non-linear baseflow occurs. These three parameters are involved in the ARNO formulation of subsurface runoff (Franchini and Pacciani, 1991; Todini, 1996), controlling the speed of baseflow release from the third soil layer (Liang et al., 1994) and, specifically, the non-linear part of the baseflow generation function. The sensitivity indices found for these parameters are consistent with the high sensitivity measures reported by Mendoza et al. (2015b), Melsen et al. (2016) and Wi et al. (2017).

The *Expt2* parameter is an exponent of the Brooks-Corey relationship (Brooks and Corey, 1964) and controls the hydraulic conductivity between the second and third soil layers. A small value for the *Expt2* parameter increases inter-layer drainage for the same soil moisture content, and therefore increases baseflow generation. The *Depth2* parameter is the thickness of the second soil layer. In general, thicker soil layers slow seasonal peak flows and increase water loss due to evapotranspiration (Xie et al., 2007). It should be noted that the parameter *Depth2* has been identified as highly sensitive by many authors (Demaria et al., 2007; Mendoza et al., 2015b; Wi et al., 2017; Gou et al., 2020; Lilhare et al., 2020; Yeste et al., 2020; Melsen and Guse, 2021). Finally, *Depth3* is the thickness of the third layer of soil, and the large sensitivities obtained here agree with the results reported by Mendoza et al. (2015b) and Wi et al. (2017).

The results in Figure 6 show that *Expt2* and *Depth2* also provide large sensitivities for metrics focused on evaporative fluxes (i.e., RR and TRANSP). Other parameters that are relevant for these processes are *LAI* and the minimum stomatal resistance (*Rmin*). Indeed, Chaney et al. (2015) reported high sensitivity of annual flow biases to variations in *Rmin*. *LAI* is a





dimensionless quantity that characterizes intra-annual variations in plant canopies, and it is defined as the one-sided green leaf area per unit ground surface area. On the other hand, *Rmin* is one of the parameters that control canopy resistance when computing transpiration from each vegetation class, following the formulations proposed by Blondin (1991) and Ducoudré et al. (1993). Both *LAI* and *Rmin* provide null sensitivities if the land cover type is bare ground; however, *Rmin* can also produce null sensitivities in vegetated grid cells.

Figure 6 reveals the large influence of hard-coded parameters on PeakSWE, SnowLegth and SUBL, in particular *NEW_ALB*, *ALB_THA* and *ALB_AA*. The *NEW_ALB* parameter is the new snow surface albedo, which controls the reflection of solar radiation and, therefore, the energy exchange between the atmosphere, forest canopy and the surface layer of the snowpack (Andreadis et al., 2009). Additionally. The *ALB_AA* and *ALB_THA* parameters represent the albedo decay in the accumulation and melting season in the snow albedo curve, respectively (USACE, 1956). These seasons are defined based on the absence

or presence of liquid water in the surface snow cover. These results correspond well with the high sensitivities reported by Mendoza et al. (2015b) for these three hard-coded parameters. Finally, the snow surface roughness length (*z0_SNOW*) also affects sublimation rates across Andean subdomains.

### 4.3 What drives parameter sensitivities across different hydroclimates?

Figure 8 shows the Spearman rank correlation coefficient, $r_s$, between parameter sensitivities and a suite of climatic,

topographic, land cover and soil-related grid cell attributes described in Table 2. The magnitude and sign of correlation quantifies how each sensitivity index varies with a given geophysical attribute. The results show that the magnitude of the correlation varies depending on the combination of metric and parameter, with the maximum correlations generally found for soil parameters, such as *DsMAX* and *Ws* with precipitation ($r_s = 0.91$) and aridity index ($r_s = -0.91$) for the FMS function (flashiness of the flow). For this example, the simulation of flow flashiness is highly sensitive to *DsMAX* and *Ws* in the wet

region but insensitive in the arid area. Conversely, the minimum correlations are found for snow parameters. It should be noted that weak correlation indicates that there is less spatial pattern in sensitivity; however, the magnitude of sensitivity index can be high or low. For example, *NEW_ALB* is a highly sensitive parameter across the domain (Figure 6).

The results in Figure 8 also indicate that high correlations (either positive or negative) are mainly associated with climate indices, which exert a stronger influence compared to the remaining attribute classes. These strong dependencies of the

parametric sensitivity on climate variables are somewhat expected, because some combinations of hydrological signatures and parameters inherit strong spatial climate patterns (Addor et al. 2018); compare, for example, aridity index Figure 1(d) with panel *DsMAX* – FMS; RMSE in Figure 7. Among climate descriptors, the aridity index, mean annual precipitation and relative humidity yield the highest correlations, and temperature exhibits a relatively lower influence on parametric sensitivity; this result that was confirmed with additional correlation analyses including only grid cells with mean annual temperatures below

5°C and 2°C (not shown). The lowest correlations are obtained for mean slope (topographic attribute), shrub fraction (land cover attribute) and mean clay content of soil (soil attribute). The key influence of climatic conditions on hydrological





behaviour is not new, since aridity is commonly regarded as the main driver of water partitioning at the land surface (Budyko, 1974; Hrachowitz et al., 2013).

Figure 8 shows that the extent to which parametric sensitivities are related to grid cell attributes depends on the target evaluation metric (i.e., runoff, evaporative processes and snow processes), with the three distinct groups containing the same influential parameters. In the following subsections, we discuss the results based on these groups, with emphasis on spatial patterns and process interpretation across our study basins. Table 6 summarizes, for each evaluation metric (i.e., physical process to be represented) and climatic zone (using the classification from Table 5), the three most important parameters. Hence, the lists contained therein can be used to guide the selection of parameters for hydrologic model calibration, based on

the hydroclimatic regime and target process that modelers would like to represent.

### 4.3.1    Runoff-oriented metrics

The results presented in Figure 7 (see RMSE and TRMSE) and Figure 8 (RMSE) show a direct relationship between the sensitivities provided by *INFILT*, and the degree of aridity, especially in semi-arid to hyper-arid subdomains. The runoff-oriented metrics are also sensitive to baseflow generation parameters *Ds*, *Ws* and *DsMAX* in most basins – with a similar spatial

distribution of $IS_j^L$ values –, excepting those located in the north and some areas in Southern Patagonia, where climatic conditions are arid or hyper-arid. In basins located in the extreme north, small sensitivities can be attributed to local climate characteristics: most precipitation events in that area occur in summer (i.e., December-March) due to orographic rains caused by air masses coming from the Amazon region, and there is usually little recharge to the aquifers. Additionally, the third soil layer in these basins generally does not reach saturation; therefore, runoff simulations in those areas are insensitive to variations

in *Ds* and *Ws* because the non-linear part of the baseflow function is only activated when the moisture storage in the third layer exceeds a threshold (Gou et al., 2020). Because of the dependency of *DsMAX* with precipitation, this parameter could be playing a key role in baseflow generation processes over Andean regions (Figure 7). Finally, a similar spatial distribution of integrated first-order sensitivities for *Ds*, *Ws* and *DsMAX* is expected, since they all focus on baseflow generation (see panels *DsMAX* - FMS, RMSE -; *Ws* -TRMSE, FMS-; *Ds* -TRMSE- in Figure 7).

*Expt2* is identified as sensitive for runoff-oriented metrics in basins with semi-arid to hyper-arid climates, characterized by small annual precipitation amounts and permanent water stress. In these hydroclimatic regimes, there is usually not enough water to reach the third soil layer, so water is stored in the second layer and drainage is mainly controlled by *Expt2*, affecting the vertical redistribution of soil moisture (FMS) and low flows (TRMSE), as shown in Figure 7. *Depth2* provides large runoff sensitivities in dry-subhumid to hyper-arid hydroclimatic regimes, for the same reasons as *Expt2*. Variations in the depth of

the second soil layer change the soil moisture of the layer, and higher (lower) values of *Depth2* for the same volumetric water content produce lower (higher) soil moisture, affecting drainage between soil layers.

Finally, *Depth3* provides large sensitivities for all runoff-oriented metrics, with similar spatial patterns to *Ws*, *Ds* and *DsMAX*, but to a smaller degree (Figure 7). *Depth3* is particularly sensitive in humid-subhumid and humid catchments, suggesting a direct relationship with mean annual precipitation, or with the size and intermittency of storms (Abdulla and Lettenmaier,





1997). In these climatic domains, periodic heavy rainfall events enable a continuous recharge of the second and third soil layers
– which may reach saturation- and thus a constant baseflow generation that affects runoff response and the retention time of
soil moisture, producing higher baseflow during wet seasons (Shi et al., 2008).

### 4.3.2    Evaporative processes

The evaluation metrics associated with these processes are RR (a measure of the overall water balance) and TRANSP (plant
transpiration), being *LAI*, *Rmin*, *Expt2* and *Depth2* the most important parameters.

Figure 7 shows a pronounced spatial variability in *LAI* sensitivities across a large domain that comprises very different land
cover types. One can note that *LAI* yields high sensitivities for nearly all hydroclimatic regimes, since this parameter controls
the evaporation from the canopy layer and canopy transpiration. In hyper-arid climates, the *LAI* is usually less important, given
the permanent water stress common for grid cells with bare soil. In summary, *LAI* is influential wherever vegetation exists,
regardless of the prevailing hydroclimatic regime.

The parameter *Rmin* yields parametric sensitivities across humid/sub-humid and humid areas (Figure 7 and Table 6). In the
canopy resistance process, there is a stomatal resistance multiplier, $g_{sm}[n]$, defined as a soil moisture stress factor that depends
on the water in the root zone for the *n*-th surface cover class. Thus, when the soil moisture in layer *n* is less than the fraction
of the moisture content at the wilting point, the value of $g_{sm}[n]$ is 0, while when the soil moisture is greater than the fractional
content of soil moisture at the critical point (~ 70% of field capacity), the value of $g_{sm}[n]$ is 1. For the intermediate condition,
$g_{sm}[n]$ values vary linearly with soil moisture in that layer, which explains why *Rmin* provides high sensitivities in very humid
(i.e., large precipitation) climates.

Finally, our results show that *Expt2* and *Depth2* yield large sensitivities to RR and TRANSP in all hydroclimatic regimes,
since they affect the soil moisture content in layer 2, which indirectly affects the $g_{sm}[n]$ factor in the canopy resistance
formulation. These parameters show a lower relative sensitivity in humid/sub-humid and humid climates, since the *Rmin*
parameter becomes more relevant when there is no soil moisture stress (i.e., $g_{sm}[n] \sim 1$).

### 4.3.3    Snow processes

Figure 7 shows that *NEW_ALB*, *ALB_THA* and *ALB_AA* yield high sensitivities throughout the study domain, especially in
areas where snow processes dominate hydrological responses. In particular, the *NEW_ALB* parameter is important throughout
the domain and reaches the highest values for snow-oriented evaluation metrics. Additionally, the results in Figure 7 show that
*ALB_AA* and *ALB_THA* dominate snow responses in different domains: the *ALB_THA* parameter yields large sensitivities in
humid and sub-humid mountain areas located southern from 34° S, with large effects on the snow season length and the
maximum SWE accumulation, while *ALB_AA* shows greater sensitivity for the other climatic regimes, affecting SnowLength
and sublimation in semiarid, colder environments in Northern Chile (26°-29° S).



## 5. Conclusions

In this study, we have re-visited parameter sensitivities in the Variable Infiltration Capacity hydrological model. To this end, we have implemented the DELSA method at every 0.05°×0.05° (~5 km) grid cell contained in 101 basins across continental Chile (i.e., a total of 5,574 grid cells), spanning a broad diversity of hydroclimatic (from hyper-arid to humid) and physiographic (e.g., topography, land cover) conditions. Our experiments consider a suite of 43 parameters included in soil, vegetation and snow process representations, with three of these corresponding to hard-coded parameters (i.e., not exposed to model users). We use eight model evaluation metrics that account for runoff components, evapotranspiration and snow processes, and conduct correlation analyses to disentangle relationships between parametric sensitivities and pixel-scale attributes. The main findings of this study are as follows:

- 31 out of 43 (i.e., 72%) parameters yield little or no sensitivity, most of which correspond to soil and vegetation processes. Therefore, calibrating such parameters will lead to minimal improvements in system representations with considerable computational costs.

- The three model evaluation metrics focused on snow accumulation and ablation processes were found to be highly sensitive to hard-coded parameters. Exposing these parameters will certainly expand our abilities to perform extensive analysis and increase our opportunities to improve model fidelity and characterize model uncertainty.

- For some evaluation metrics, the climate attributes examined here are highly correlated with parameter sensitivities, which therefore inherit spatial patterns observed in climate variables across the territory. In particular, mean annual precipitation and the aridity index are highly correlated with *Ds*, *Ws* and *DsMAX* sensitivities when examining RMSE, TRMSE and FMS. Unexpectedly, temperature yields a relatively lower influence among climates descriptors, even for metrics and parameters associated with snow processes. The rest of the attributes (topographic, soil and land cover) provided generally low correlations, and therefore small predictive power on parameter sensitivities.

- Parametric sensitivities are strongly related with the climate types in the case study basins. In humid environments, the most important parameters are related to the third soil layer (*Ws*; *Ds*; *DsMAX* and *Depth3*) and vegetation (*Rmin*); in arid regimes, the most influential parameters are associated with the firsts soil layers (*INFILT*; *Expt2*; and *Depth2*).

- In snow-dominated areas, the hard-coded parameters *NEW_ALB*; *ALB_THA* and *ALB_AA* provide large sensitivities to maximum SWE, snow season length and sublimation.

- The leaf area index (*LAI*) is a crucial parameter wherever there is vegetation on the ground, especially if it presents an overstory. Although such condition is more frequent in humid environments, the relevance of this parameter depends on vegetation characteristics rather than the underlying climatic conditions.

Overall, our study contributes to the existing literature by providing guidance on relevant VIC parameters for a suite of target processes and climate types, using a large number of modelling units at a relatively high (~5 km) spatial resolution. Future studies aiming at improving spatial calibration density and/or parameter regionalization techniques using VIC – or any similar hydrology or land surface model – could incorporate this information to define spatially varying target parameters, and examine

to what extent the spatial patterns in parameter sensitivities relate to calibrated parameter fields. Finally, the strong correlations found here between parameter sensitivities and hydroclimatic properties reaffirm the need to incorporate periods with

contrasting climate characteristics in sensitivity analysis and calibration strategies in order to achieve more credible hydrologic model simulations under changing climatic conditions.

**Data availability**

Land cover descriptors for all grid cells and reference VIC model outputs used to compute performance metrics were obtained from the National Water Balance database (DGA, 2018, 2019, 2020). This information may be requested through the website

https://siac.mop.gob.cl/. Other grid cell attributes were obtained from the United States Geological Survey dataset (https://earthexplorer.usgs.gov/), the CR2met dataset (https://www.cr2.cl/), and SoilGrids250m 2.0 (https://soilgrids.org/). Catchment scale hydrometeorological data were obtained from the CAMELS-CL dataset (Alvarez-Garreton et al., 2018).

**Author contributions**

All the authors were involved in the conceptualization of this study. US and PM designed the methodology and analysis

framework and drafted the paper. US configured the VIC model, conducted simulations, analysed the results and created all the figures. AN and NM provided insights into the sensitivity analysis results. All the authors discussed the results and contributed to writing, reviewing and editing the manuscript.

**Competing interests**

The authors declare that they have no conflict of interest.

**Acknowledgments**

Pablo A. Mendoza received support from Fondecyt Project 11200142 and CONICYT/PIA Project AFB180004. This research was partially supported by the supercomputing infrastructure of the NLHPC (ECM-02). The National Center for Atmospheric Research is a major facility sponsored by the National Science Foundation under Cooperative Agreement No. 1852977. We thank Eduardo Muñoz-Castro and Nicolás Vásquez for their advice and assistance in setting up model simulations, and Ximena

Vargas and Miguel Lagos for their suggestions on earlier versions of this manuscript.

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





**Table 1. Summary of sensitivity analysis studies conducted with VIC, that incorporate at least five parameters[a].**

| Study | Region | Number of sites or catchments | Target variables or metrics | Number of parameters included in SA | Methods | Most sensitive parameters |
|---|---|---|---|---|---|---|
| Liang & Guo (2003) | Red-Arkansas River basin, USA | 3 sites | Annual runoff, annual ET, annual mean soil moisture, and annual mean sensible heat flux | 5 pre-defined parameters | Fractional Factorial Analysis (FFA) | Varied with site characteristics |
| Demaria et al. (2007) | South-East of the United States | 4 catchments | Five metrics (including RMSE, bias and correlation) formulated with daily streamflow and daily baseflow | 10 soil parameters | Regional Sensitivity Analysis | *INFILT*, *Expt_i* and *Depth2* |
| Chaney et al. (2015) | Global | 1° resolution grid cells over the globe excluding Greenland and Antarctica (15,836 grid cells in total). | Annual flow biases, runoff seasonality and daily flow percentiles. | 8 soil and one vegetation parameter | Distance between a priori CDF and behavioral parameter CDF | *INFILT*, *DsMAX*, *Expt_i*, and *Rmin* for annual biases; *INFILT* and *DsMAX* when adding the monthly constraint; *INFILT*, *DsMAX*, *Expt_i*, and *Rmin* dominate daily flow extremes. |
| Mendoza et al. (2015b) | Colorado Headwaters Region, USA | 3 headwater basins | RMSE(Q) with Q at daily time steps | 34 soil, vegetation and snow parameters | DELSA | *INFILT*, *Ds*, *DsMAX*, *Ws*, *Depth2*, *Depth3*, *NEW_ALB*, *ALB_AA*, *ALB_THA* |
| Melsen et al. (2016) | Thur basin, Switzerland | 3 catchments | NSE(Q), KGE(Q) and KGE(log(Q)) with Q at daily and hourly time steps | 28 soil, vegetation and snow parameters | DELSA | *INFILT*, *Ds*, *Expt2*, *DsMAX* |
| Wi et al. (2017)[b] | American River basin, USA | One catchment (North Fork sub-basin) | NSE(Q) with Q at daily time steps | 15 soil, snow and routing parameters | Morris | *INFILT*, *Ds*, *DsMAX*, Ws, *Depth1*, *Depth2*, *Depth3*, snow *Tmax* & *Tmin* |
| Bennett et al. (2018) | Colorado River basin, USA | 7 grid cells across the basin, with ~7 km horizontal resolution | Projected changes (i.e., 2070-2099 minus 1970-1999 averages) in mean annual runoff, June soil moisture, March SWE and annual ET | 46 soil and vegetation parameters | Variance-based Sensitivity Analysis, applied to a Gaussian process emulator developed for VIC at each grid cell | *DsMAX*, *Ds* and *Depth3* for projected changes in runoff, ET and soil moisture; wintertime canopy fraction and wintertime *LAI* for projected changes in SWE |



| Study | Region | Number of sites or catchments | Target variables or metrics | Number of parameters included in SA | Methods | Most sensitive parameters |
|---|---|---|---|---|---|---|
| Gou et al. (2020) | 10 major river basins in China | 14 catchments | NSE(Q) with Q at monthly time steps | 13 soil parameters | Three qualitative (SOT, MARS, DT) and one quantitative (Sobol') method | *INFILT*, *Depth1* and *Depth2* are the overall most influential parameters on streamflow |
| Lilhare et al. (2020) | Lower Nelson River basin, Canada | 10 sub-basins | NSE(Q), KGE(Q) and PBIAS(Q) with Q at daily time steps | 6 soil parameters | VARS | *INFILT* and *Depth2* arised as the most sensitive parameters. Relative importance depends on the catchment. |
| Yeste et al. (2020) | Duero River basin, Iberian Peninsula | 31 headwater basins | Temporal averages of surface runoff, baseflow, total runoff, ET, and total soil moisture | 5 soil parameters | SRC | Surface runoff, baseflow, total runoff, ET and SM1 are mainly sensitive to *INFILT* and *Depth2*. SM2 affected mostly by *Depth2*, and SM3 affected by *Ds*, *DsMAX* and *Ws* |
| Melsen and Guse (2021) | Contiguous United States (CONUS) | 605 catchments | Mean simulated streamflow and the day of the year when half of the streamflow volume has passed (i.e., streamflow timing) | 17 soil, vegetation and snow parameters | DELSA | *Rmin*, *Depth2* and *Expt2* are the most sensitive parameter for mean annual discharge; *Ds*, *DsMAX*, *Depth2*, *Rmin*, *Expt2* and *Depth3* control streamflow timing |
| This study | Continental Chile | 5,574 grid cells (0.05°×0.05°) across 101 basins | 8 metrics computed at daily time steps | 43 soil, vegetation and snow parameters | DELSA | *INFILT*, *Ds*, *DsMAX*, *Ws*, *Expt2*, *Depth2*, *Depth3*, *Rmin*, *LAI*, *NEW_ALB*, *ALB_THA*, *ALB_AA* |

[a]The studies are listed in order of publication date. The present study has been added for completeness.
[b]We exclude two routing parameters that were found sensitive, but were not used in the other studies.
FFA: Factorial Fractional Analysis (Montgomery, 1991)
DELSA: Distributed Evaluation of Local Sensitivity Analysis (Rakovec et al., 2014).
DT: Delta test (Pi and Peterson, 1994).
SOT: Sum-Of-Trees model (Chipman et al., 2010).
MARS: Multivariate Adaptive Regression Splines (Friedman, 1991).
VARS: Variariogram Analysis of Response Surfaces (Razavi and Gupta, 2016).
SRC: Standardized Regression Coefficients (Saltelli et al., 2008).
VISCOUS: VarIance-based Sensitivity analysis using COp- UlaS (Sheikholeslami et al., 2021).



**Table 2. List of physiographic and hydroclimatic attributes used to characterize model grid cells.**

| Predictor | Class | Description | Data source |
|---|---|---|---|
| Elevation | Topographic | Mean elevation (m.a.s.l.) | DGA (2018, 2019, 2020) |
| Slope | Topographic | Mean topographic slope (°) | Digital Elevation SRTM 1 Arc-Second Global (https://earthexplorer.usgs.gov/) |
| Precipitation | Climate | Mean annual precipitation (mm/yr) | CR2MET (Boisier et al., 2018) (https://www.cr2.cl/) |
| Temperature | Climate | Mean temperature (°C) | CR2MET (Boisier et al., 2018) (https://www.cr2.cl/) |
| Humidity | Climate | Mean relative humidity (-) | CR2MET (Boisier et al., 2018) (https://www.cr2.cl/) |
| Aridity | Climate | Aridity index (-), ratio of long-term potential evaporation to precipitation | - |
| Clay | Soil | Soil clay content (%) average over all layers | SoilGrids250m (Hengl et al., 2017) (https://soilgrids.org/) |
| Bare soil | Land cover | Fraction of bare soil | DGA (2018, 2019, 2020) |
| Forest | Land cover | Fraction of forest | DGA (2018, 2019, 2020) |
| Grasslands | Land cover | Fraction of grasslands | DGA (2018, 2019, 2020) |
| Shrub | Land cover | Fraction of shrub | DGA (2018, 2019, 2020) |
| Snow | Land cover | Fraction of snow cover | DGA (2018, 2019, 2020) |


**Table 3. Parameters of the VIC model considered in this study.**

| Parameter | Description | Units | Min | Max | Comment |
|---|---|---|---|---|---|
| Soil parameters | | | | | |
| INFILT | Variable infiltration curve parameter | - | 0.001 | 0.4 | Based on Mendoza et al. (2015b) |
| Ds MAX | Maximum velocity of baseflow | mm/d | 1 | 50 | Based on Melsen et al., (2016) |
| Ds | Fraction of $Ds_{max}$ where non-linear baseflow occurs | - | 0.00005 | 1 | Based on Mendoza et al. (2015b) |
| Ws | Fraction of maximum soil moisture where non-linear baseflow occurs | - | 0.0009 | 1 | Based on Mendoza et al. (2015b) |
| c | Exponent used in baseflow curve | - | 1 | 4 | Based on Melsen et al. (2016) |
| $Expt_i$ | Exponent in Campbell's equation for hydraulic conductivity of soil layer i | - | 5 | 30 | Based on Melsen et al. (2016) |
| $Ksat_i$ | Saturated hydraulic conductivity of soil layer i | mm/d | 1 | 10000 | Based on Demaria et al. (2007) |
| $Depth_1$ | Thickness of layer 1 (uppermost) | m | 0.01 | 0.5 | Based on Demaria et al. (2007) |
| $Depth_2$ | Thickness of layer 2 | m | $Depth_1 + 0.1$ | $Depth_1 + 4$ | Based on Melsen et al. (2016) |
| $Depth_3$ | Thickness of layer 3 (lowermost) | m | 0.1 | 4 | Based on Melsen et al. (2016) |
| dp | Soil thermal damping depth | m | 1 | 3.75 | Based on Gates & Evans (1964) and Al |



| Parameter | Description | Units | Min | Max | Comment |
|-----------|-------------|-------|-----|-----|---------|
| | | | | | Nakshabandi & Kohnke (1965) |
| quartz$_i$ | Quartz content of soil layer i | - | 0.1 | 0.82 | Based on Hogue et al. (2005) and Rosero et al. (2010) |
| bulk density$_i$ | Bulk density of layer i | kg/m$^3$ | 1200 | 1609 | Based on Cosby et al. (1984); Rawls et al. (1992) and Reynolds et al. (2000) |
| rough | Surface roughness of bare soil | m | 0.0001 | 0.08 | Based on Woodward (1999) |
| Resid moist | Residual soil moisture of layer i | - | 0.02 | 0.109 | Based on Rawls et al. (1992) |
| Vegetation parameters | | | | | |
| rarc | Architectural resistance of vegetation type | s/m | 2 | 50 | Based on Ducoudré et al. (1993) |
| Rmin | Minimum stomatal resistance of vegetation type | s/m | 30 | 300 | Based on Melsen et al. (2016) |
| LAI* | Leaf-area index of vegetation type | - | **0.1** | **1.16** | Multipliers obtained from leaf-area index range 0.01-7 based on Dorman & Sellers (1989) and Myneni et al. (1997) |
| ALB* | Shortwave albedo for vegetation type | - | **1** | **1.65** | Multiplier obtained from shortwave albedo range 0.1-0.33 based on Dorman & Sellers (1989) |
| ROU* | Vegetation roughness length | - | **0.82** | **2.11** | Multiplier obtained from roughness length range 0.06-2.6 m based on Dorman & Sellers (1989) |
| DIS* | Vegetation displacement height | - | **0.82** | **2.11** | Multiplier obtained from roughness length range 0.06-2.6 m based on Dorman & Sellers (1989) and VIC definitions |
| Root depth i | Root zone thickness (sum of depths is total depth of root penetration) of layer i | m | 0.1 | 3 | Based on Melsen et al. (2016) |
| Root fraction i | Fraction of root in the current root zone of layer i. | - | 0 | 1 | Based on Bohn & Vivoni (2016) |
| Snow and general parameters | | | | | |
| z0 SNOW | Surface roughness of snowpack | m | 0.0001 | 0.01 | Based on range suggested by Marks & Dozier (1992) and Reba et al. (2014) |
| Tmin | Minimum temperature at which rain can fall. | °C | -1.5 | 0 | Based on Melsen et al. (2016) |





| Parameter | Description | Units | Min | Max | Comment |
|---|---|---|---|---|---|
| Tmax | Maximum temperature at which snow can fall. | °C | Tmin + 0.5 | Tmin + 1.5 | Based on Melsen et al. (2016) |
| NEW ALB | New snow albedo | - | 0.7 | 0.99 | Based on Mendoza et al. (2015b) |
| ALB AA | Base in snow albedo function (accumulation) | - | 0.88 | 0.99 | Based on Mendoza et al. (2015b) |
| ALB THA | Base in snow albedo function (melt) | - | 0.66 | 0.98 | Based on Mendoza et al. (2015b) |

*This parameter is temporally distributed (monthly variations) and, therefore, its sensitivity is analyzed based on multipliers. Although description and units refer to actual parameters of VIC, parameter values in bold represent the multiplier values (instead of actual parameters).


**Table 4. Parameter sensitivity metrics used in this study.**

| Notations | Short descriptions | Formulas | Indicator of processes |
|---|---|---|---|
| RMSE | Root-mean-squared-error | $\sqrt{\dfrac{1}{N}\sum_{t=1}^{N}\left(Q_t^{sim}-Q_t^{ref}\right)^2}$ | High flows |
| TRMSE | Transformed-root-mean-squared-error | $\sqrt{\dfrac{1}{N}\sum_{t=1}^{N}\left(Z_t^{sim}-Z_t^{ref}\right)^2}$ $Z_t=\dfrac{(1+Q_t)^\lambda-1}{\lambda};\ \lambda=0.3$ | Low flows |
| FMS | Flow duration curve midsegment slope difference | $\left\|\dfrac{Q_{m_1}^{sim}-Q_{m_2}^{ref}}{m_1-m_2}-\dfrac{Q_{m_1}^{obs}-Q_{m_2}^{ref}}{m_1-m_2}\right\|$ | Variability, or flashiness, of the flow magnitudes |
| RR | Runnof ratio difference | $\left\|\dfrac{R_{sim}}{P_{sim}}-\dfrac{R_{ref}}{P_{ref}}\right\|$ | Overall water balance (ET processes) |
| PeakSWE | Peak SWE difference | $\left\|max\{SWE_t\}_{sim}-max\{SWE_t\}_{ref}\right\|$ | Maximum long-term SWE accumulation |
| SnowLength | Snow length difference | $\left\|\sum days \in SWE_{sim}>x-\sum days \in SWE_{ref}>x\right\|$ $x=1\ mm$ | The number of days when snow is on the ground |
| SUBL | Sublimation difference | $\left\|Subl_{sim}-Subl_{ref}\right\|$ | Mean error on sublimation estimation |
| TRANSP | Transpiration difference | $\left\|Transp_{sim}-Transp_{ref}\right\|$ | Mean error on transpiration estimation |

N, number of time steps; $Q_t$, flow for time step t; $Z_t$, flow transformed for time step t; $Q_{m_1}$, $m_1$ percentile flow of simulated flow duration curve; $m_1=70$; $Q_{m_2}$, $m_2$ percentile flow of simulated flow duration curve; $m_2=30$; R, grid-averaged mean annual runnof; P, grid-averaged mean annual precipitation; $SWE_t$, Snow water equivalent for time step t; Subl, grid-averaged mean annual sublimation; Transp, grid-averaged mean annual transpiration.






**Table 5. Climate classification used to group model grid cells.**

| Classification | Humid | Humid sub-humid | Dry sub-humid | Semi-arid | Arid | Hyper arid |
|---|---|---|---|---|---|---|
| Aridity index | <1 | 1 to 1.53 | 1.53 to 2 | 2 to 5 | 5 to 20 | >20 |
| Number of grid cells | 2189 | 772 | 318 | 992 | 803 | 499 |

**Table 6. Summary with the most sensitive VIC parameters found for each metric (rows) and climatic type. The three most important parameters are determined based on the median of integrated first-order DELSA sensitivity indices and are sorted by ranking (i.e., 1st, 2nd, and 3rd most sensitive).**

| Classification | Humid | Humid sub-humid | Dry sub-humid | Semi-arid | Arid | Hyper arid |
|---|---|---|---|---|---|---|
| RMSE | Ds MAX<br>Ws<br>Depth 3 | Ds MAX<br>Ws<br>Depth 3 | INFILT<br>Ws<br>Ds MAX | INFILT<br>Expt 2<br>Depth 2 | INFILT<br>Expt 2<br>Depth 2 | INFILT<br>Expt 2<br>Depth 2 |
| TRMSE | Ds MAX<br>Ws<br>Ds | LAI<br>Ds MAX<br>Ds | LAI<br>Depth 2<br>Ds | INFILT<br>Depth 2<br>Expt 2 | INFILT<br>Expt 2<br>Depth 2 | INFILT<br>Expt 2<br>Depth 2 |
| FMS | Ds MAX<br>Ds<br>Ws | Ds<br>Ds MAX<br>LAI | Ds<br>Ds MAX<br>LAI | Ds<br>Ds MAX<br>Depth 2 | Depth 2<br>Expt 2<br>Ds | Depth 2<br>Expt 2<br>Ds |
| RR | LAI<br>Rmin<br>Expt 2 | LAI<br>Rmin<br>Expt 2 | LAI<br>Expt 2<br>Depth 2 | LAI<br>Expt 2<br>Depth 2 | INFILT<br>Depth 1<br>Expt 2 | Depth 2<br>INFILT<br>Depth 1 |
| PeakSWE | NEW ALB<br>ALB THA<br>LAI | NEW ALB<br>ALB THA<br>ALB AA | NEW ALB<br>z0 SNOW<br>ALB THA | NEW ALB<br>ALB AA<br>T max | NEW ALB<br>ALB AA<br>z0 SNOW | NEW ALB<br>ALB AA<br>z0 SNOW |
| SnowLength | NEW ALB<br>ALB THA<br>ALB AA | NEW ALB<br>ALB AA<br>ALB THA | NEW ALB<br>ALB AA<br>ALB THA | NEW ALB<br>ALB AA<br>T max | NEW ALB<br>ALB AA<br>ALB THA | NEW ALB<br>ALB AA<br>z0 SNOW |
| SUBL | z0 SNOW<br>NEW ALB<br>ALB THA | NEW ALB<br>z0 SNOW<br>T max | z0 SNOW<br>NEW ALB<br>T max | NEW ALB<br>z0 SNOW<br>ALB AA | NEW ALB<br>z0 SNOW<br>ALB AA | NEW ALB<br>ALB AA<br>z0 SNOW |
| TRANSP | LAI<br>Rmin<br>Expt 2 | LAI<br>Rmin<br>Depth 2 | LAI<br>Depth 2<br>Expt 2 | LAI<br>Depth 2<br>Expt 2 | Expt 2<br>LAI<br>Depth 2 | Expt 2<br>Depth 2<br>LAI |






**Figure 1. Spatial distribution of climatic and physiographic attributes across all grid cells: (a) mean annual precipitation (period 1979 – 2020), (b) mean annual temperature (period 1979 – 2020), (c) mean elevation, (d) aridity index and (e) bare soil fraction. In each panel, the back thick lines represent the boundaries of six basins representative of the hydroclimatic diversity within the study domain, from north to south: (a) Loa River upstream Lequena reservoir; (b) Pulido River at Vertedero; (c) Colorado River before junction with Maipo River; (d) Palos River at junction with Colorado River; (e) Biobío River at Rucalhue; (f) Cautin River at Cajon.**





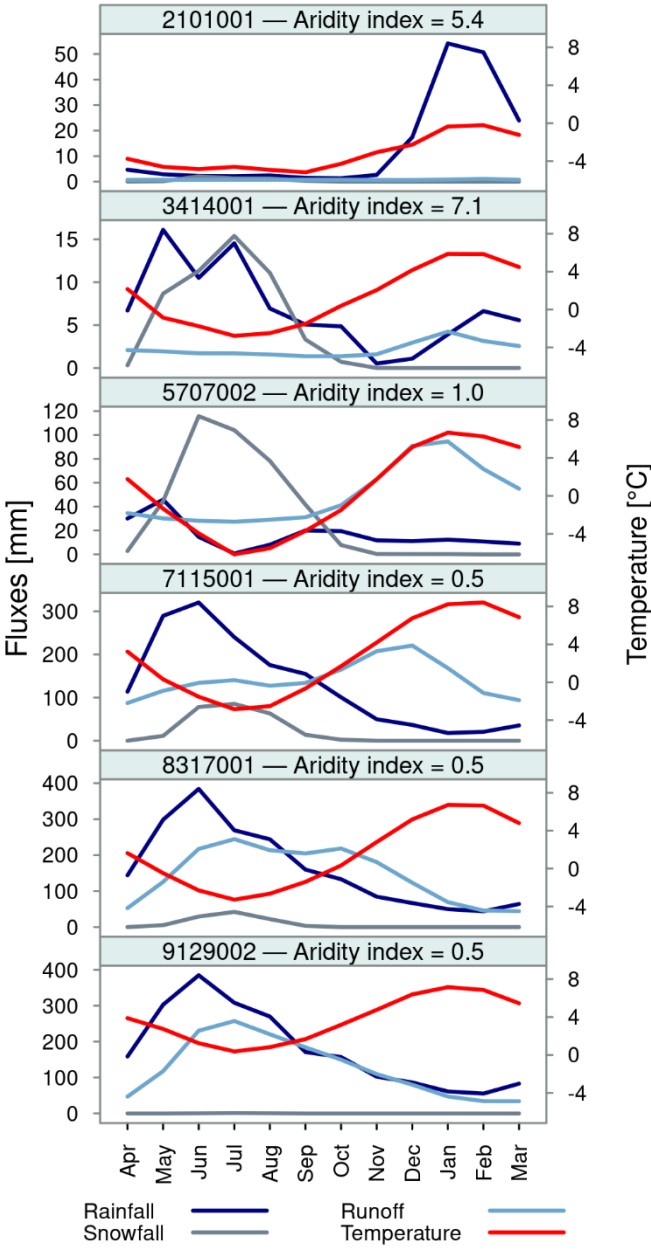

**Figure 2**. **Seasonal cycles of catchment-averaged precipitation, runoff and temperature (period 1981 – 2018) for six basins representative of the hydroclimatic diversity within the study domain. From north to south: (a) Loa River upstream Lequena reservoir; (b) Pulido River at Vertedero; (c) Colorado River before junction with Maipo River; (d) Palos River at junction with Colorado River; (e) Biobío River at Rucalhue; (f) Cautín River at Cajon. The location of these catchments is shown in Figure 1.**







**Figure 3. Comparison of cumulative frequency distributions of first-order DELSA indices ($S_j^L$) across six hydroclimatically different basins (displayed in different rows). The location and seasonal cycles for these basins are displayed in Figures 1 and 2, respectively. Results are displayed for the most sensitive parameter associated with each evaluation metric (displayed in different columns), so each panel (excepting those in the last row) comprises the CDFs of all grid cells contained in a specific basin, for a particular combination of metric/parameter. The most sensitive parameter was determined based on the median sensitivity index $IS_j^L$ from all the grid cells contained in the study domain (see text for details). The number next to each basin code at the top of this figure is the catchment-scale aridity index.**




**Figure 4. Spatial distribution of integrated first-order DELSA sensitivity indices ($IS_j^l$) for the leaf area index (*LAI*) across a humid subdomain located in Southern Chile. Results are displayed for eight sensitivity metrics: (a) RMSE, (b) TRMSE; (c) FMS; (d) RR; (e) PeakSWE; (f) SnowLength; (g) SUBL; (h) TRANSP.**





**Figure 5. Same as in Figure 4, but for the base in the snow albedo function *ALBTHA* (melt).**




**Figure 6.** Boxplots comprising integrated first-order DELSA sensitivity indices ($IS_j^L$) from all modeling units (5,574 grid cells). Results are displayed for all parameters (x-axis) and sensitivity metrics which are presented in different panels: (a) RMSE, (b) TRMSE; (c) FMS; (d) RR; (e) PeakSWE; (f) SnowLength; (g) SUBL; (h) TRANSP.




**Figure 7**. **Integrated first-order DELSA sensitivity indices for all grid cells within our study basins. The results are displayed only the 12 most sensitive parameters, and their associated most impacted metrics.**




**Figure 7. (continued).**



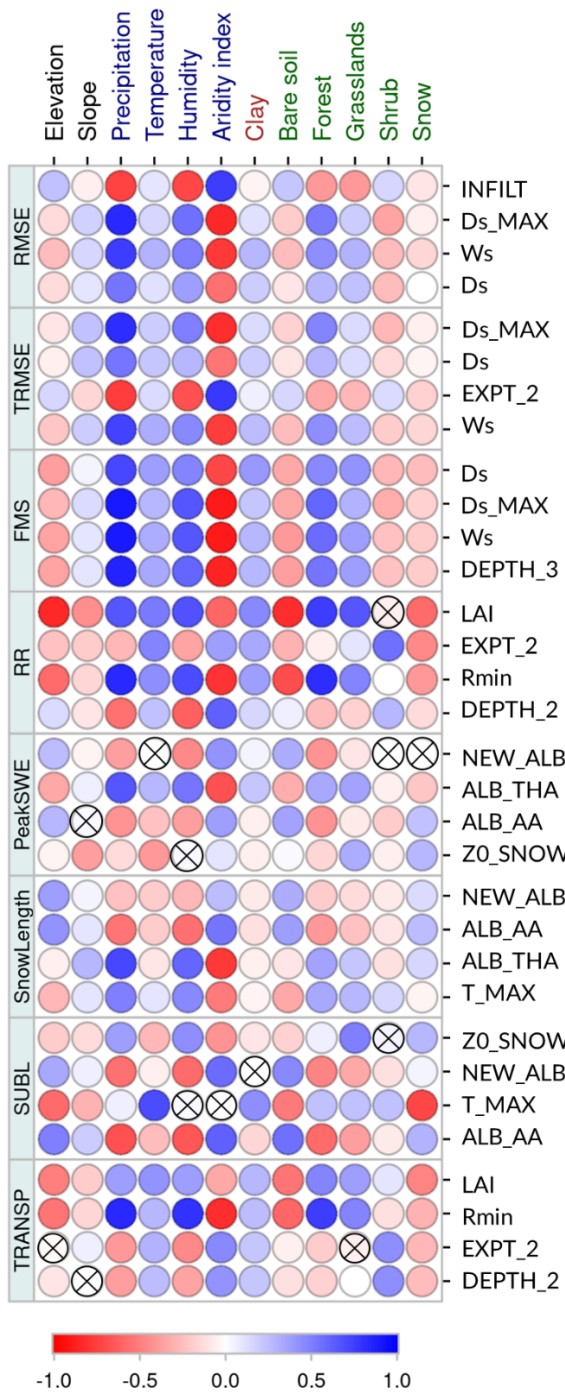

**Figure 8**. **Spearman rank correlation coefficient between integrated first-order DELSA sensitivity indices $IS_j^L$ and**
**grid cell characteristics. Results are displayed only for the four most sensitive parameters affecting each metric. The**
**crosses indicate correlation values whose p-values are lower than 0.05.**