# Peer review of "Revisiting parameter sensitivities in the Variable Infiltration Capacity model across a hydroclimatic gradient"

_Hydrology and Earth System Sciences, 2021_

## Author Comment (AC1)

Replies to reviews

**"Revisiting parameter sensitivities in the Variable Infiltration Capacity model across a hydroclimatic gradient"**

Ulises M. Sepúlveda, Pablo A. Mendoza, Naoki Mizukami and Andrew J. Newman

We thank the Editor and the reviewers for their time in commenting on our paper. We provide responses to each individual point below. For clarity, comments are given in italics, and our responses are given in plain blue text.

**Anonymous Referee #1**

*This paper studies the sensitivity of the VIC model to 43 soil, vegetation, and snow parameters using the DELSA sensitivity analysis method in about 5500, 0.05 degree grid cells in Chile. The authors find that different goodness-of-fit metrics (the authors try eight of these) have more or less sensitivity to different parameters. Precipitation and aridity are also found to control parameter sensitivity. Recommendations are provided for how to select calibration parameters based on climatology and which variable is of interest.*

*Summary of findings:*

- *VIC is overparameterized - only 12 of the 43 parameters are sensitive – 7 soil parameters, 2 vegetation parameters, and 3 snow parameters.*
- *Mean annual precipitation and aridity control which parameters are sensitive.*
- *Leaf area index and hard-coded snow parameters are sensitive.*
- *Provides guidance on the most relevant parameters for model calibration depending on the target process (runoff, snow, or ET) and climate type (humid/arid).*

*This paper is well-written, and I think it will be a useful resource for VIC modelers, as it describes the VIC parameters in depth, where they came from, and what good min/max values are for calibration. Its point that some of the hard-coded snow model parameters are sensitive and perhaps should be exposed to users is well-taken.*

We thank the reviewer for his/her positive feedback.

*My main criticism of this paper is that some of the findings, e.g. that VIC is overparameterized and certain parameters are more sensitive under certain conditions – are well-known from other studies (such as Demaria et al., 2007 and Gou et al., 2020).*

We agree that our study confirms that VIC is overparameterized, and that relative sensitivities vary depending on the specific metrics analysed. Nevertheless, our results are not directly comparable to the studies mentioned by the reviewer, since Demaria et al. (2007) and Gou et al. (2020) only included soil parameters – 7 and 15, respectively – in their analyses, ignoring the potential effects that a large number of vegetation parameters and hard-coded snow parameters exert on simulated hydrological responses.

More generally, the results and conclusions reported in this paper are not directly comparable to any study listed in Table 1, due to differences in the experimental design

and domains of interest. In particular, our study considers (1) a large number of soil, vegetation and snow parameters – only comparable to Bennett et al. (2018), who included 46 parameters (though excluding snow) –, (2) a larger number of process-oriented metrics (compared to the previous efforts listed in Table 1), and (3) a very large sample of grid cells at a relatively high (~5 km) horizontal resolution, spanning very different physiographic characteristics and hydroclimatic conditions. Hence, our study contributes to the existing literature by providing guidance on relevant VIC parameters for a suite of target processes and climate types. We have strengthened these points by adding the following paragraph in section 5 (L450-459):

"This study reaffirms overparameterization issues in the VIC model (Demaria et al., 2007; Gou et al., 2020), and that relative parameter importance varies depending on the specific metric or variable analysed (Chaney et al., 2015; Bennett et al., 2018; Yeste et al., 2020; Melsen & Guse, 2021), and physiographic or climatic site characteristics (Liang & Guo, 2003; Demaria et al., 2007; Lilhare et al., 2020). However, the results and conclusions reported here are not directly comparable to previous studies due to differences in the experimental designs and domains of interest. In particular, our study considers (1) a large number of soil, vegetation and snow parameters – only comparable to Bennett et al. (2018), who included 46 parameters (excluding snow processes) over the semiarid Colorado River basin –, (2) a larger number of process-oriented metrics (compared previous efforts listed in Table 1), and (3) a very large sample of grid cells at a relatively high (~5 km) horizontal resolution, spanning very different physiographic characteristics and hydroclimatic conditions. Hence, our study contributes to the existing literature by providing guidance on relevant VIC parameters for a suite of target processes and climate types."

*On the other hand, examining parameter interactions, which the authors say is possible using the DELSA method, might be more interesting.*

In this paper, we used the same DELSA formulation proposed in the original paper (Rakovec et al., 2014), which does not enable the quantification of parameter interactions. The latter would require mathematical developments that are out of the scope of this paper. We have re-worded the text to clarify this (L191-194):

"Although DELSA only examines first-order sensitivities, it has unexplored potential to be expanded with the aim to quantify parameter interactions (Zegers et al., 2020), which could be achieved by including additional terms in the local total variance (Sobol' & Kucherenko, 2010)."

*Some other critiques:*

1) *I wonder whether the amount of peak SWE is a useful goodness-of-fit parameter. I imagine that many combinations of parameters could give the same peak SWE, since it is an integrated measure of the entire season's snowfall. This would be worth mentioning in Section 3.4 performance metrics.*

Characterising parameter equifinality, which also occurs in other metrics, is beyond the scope of this study. Nevertheless, we clarify what peak SWE represents following the reviewer's suggestion (L234-236):

"We use two metrics to characterize snow cover processes: the difference in long-term simulated peak SWE – an integrated measure of processes occurring during the snowfall season –, and the difference in snow cover duration, quantified by the number of days with snow on the ground (Mizukami et al., 2014)."

2) *You study 101 catchments throughout Chile, but only a few catchments are highlighted in the figures. Is there any justification for why these catchments are spotlighted?*

We highlight a small sample of six basins in Figures 1-3 because they have different aridity indices and seasonal cycles of rainfall, snowfall, runoff and temperature, illustrating the diversity of hydroclimate regimes across the country. These catchments exemplify the transition from an arid, rainfall dominated regime (Far North), to semi-arid, snow-dominated basins, towards mixed and rainfall dominated streamflow regimes with lower aridity indices in southern Chile (as described in L120-124).

We have added the following text to clarify the motivation of highlighting a small sample of six basins (L118-120):

"Figure 2 illustrates this by showing the aridity indices and seasonal cycles of rainfall, snowfall, runoff and temperature for a sample of six basins with very different hydroclimatic regimes."

3) *In Figure 3, indicate whether the rows are organized by latitude (they appear to be, but it would help readers interpret the figure if this were more clear).*

We have included "(displayed in different rows, sorted by latitude)", following the reviewer's recommendation.

4) *L315: SnowLegth typo*

Thanks for noting this. We have corrected the typo accordingly.

5) *In Section 4.3.1, it is claimed that humid environments enable recharge of the lower soil layers and thus cause increased sensitivity of baseflow parameters. Is this true for all wet catchments, or does it depend whether precipitation occurs as rainfall or snowfall? I imagine that snowy catchments will have more sensitivity to baseflow parameters, as water will have more time to infiltrate into the soils. (I'm assuming here that snowmelt runoff is generated more gradually than rainfall runoff.)*

We thank the reviewer for this insightful observation. The climatic classification of grid cells used here – spanning from humid to hyper-arid – is solely based on the mean annual aridity index (Table 5) and, therefore, humid grid cells include both rainfall and snowfall dominated climates. To test the reviewer's hypothesis, we computed the fraction of precipitation falling as snowfall ($f_s$) in the 2,189 grid cells classified as 'humid', defining the modeling units where $f_s \geq 0.15$ as snowfall-dominated

(Berghuijs et al., 2014). We estimate $f_s$ with the 3-hourly meteorological time series used to compute parameter sensitivities (April/2001 – March/2011), using a temperature threshold of 1 °C for partitioning precipitation between rainfall and snowfall (Hock, 2003). As a result, we find that 711 (32.5%) humid grid cells are snowfall dominated, and the remaining 1,478 cells (67.5%) are rainfall dominated.

The stratification of parameter sensitivities in humid grid cells based on the dominant precipitation phase reveals that the three most important parameters in rainfall dominated grid cells remain the same for all metrics. In snowfall dominated grid cells, the ranking of relevant parameters is the same for all metrics, excepting RMSE and TRMSE (Table 1), for which ALB THA arises as the most important parameter. Such behavior is expected because ALB THA affects snow albedo during melting periods; nevertheless, the baseflow parameters Ds Max and Ds are still important in humid, snowfall dominated grid cells, with $IS_L$ medians that are similar to those obtained from all humid grid cells. Therefore, we conclude that baseflow parameters provide important sensitivities across humid grid cells, regardless of the dominant precipitation phase.

We have included the following text in section 4.3.1 to make this point (L390-392): "Interestingly, in humid subdomains baseflow parameters yield high sensitivities in both rainfall and snowfall dominated grid cells, although ALB THA emerges as the most important parameter for RMSE and TRMSE in snowfall dominated grid cells (not shown)."

Table 1: List with the three most sensitive parameters for all humid grid cells (same as in Table 6), and for snowfall dominated humid grid cells

| Metric | All humid grid cells (n = 2,189; same as in Table 6) | | Snowfall dominated humid grid cells (n = 711) | |
|---|---|---|---|---|
| | Parameter | $IS_l$ median | Parameter | $IS_l$ median |
| RMSE | Ds Max | 0.16 | ALB THA | 0.211 |
| | Ws | 0.129 | Ds Max | 0.163 |
| | Depth 3 | 0.109 | Ws | 0.133 |
| | | | | |
| TRMSE | Ds Max | 0.152 | ALB THA | 0.202 |
| | Ws | 0.11 | Ds Max | 0.159 |
| | Ds | 0.104 | Ws | 0.118 |

*6)  L380: syntax should be "LAI, Rmin, etc. being the most important parameters."*

Thanks for noting this. We have corrected the syntax following the reviewer's recommendation.

References

Bennett, K. E., Urrego Blanco, J. R., Jonko, A., Bohn, T. J., Atchley, A. L., Urban, N. M., & Middleton, R. S. (2018). Global Sensitivity of Simulated Water Balance Indicators Under Future Climate Change in the Colorado Basin. *Water Resources Research*, *54*(1), 132–149. https://doi.org/10.1002/2017WR020471

Berghuijs, W. R., Woods, R. A., & Hrachowitz, M. (2014). A precipitation shift from snow towards rain leads to a decrease in streamflow. *Nature Climate Change*, *4*(7), 583–586. https://doi.org/10.1038/nclimate2246

Chaney, N. W., Herman, J. D., Reed, P. M., & Wood, E. F. (2015). Flood and drought hydrologic monitoring: The role of model parameter uncertainty. *Hydrology and Earth System Sciences*, *19*(7), 3239–3251. https://doi.org/10.5194/hess-19-3239-2015

Demaria, E. M., Nijssen, B., & Wagener, T. (2007). Monte Carlo sensitivity analysis of land surface parameters using the Variable Infiltration Capacity model. *Journal of Geophysical Research Atmospheres*, *112*(11), 1–15. https://doi.org/10.1029/2006JD007534

Gou, J., Miao, C., Duan, Q., Tang, Q., Di, Z., Liao, W., et al. (2020). Sensitivity Analysis-Based Automatic Parameter Calibration of the VIC Model for Streamflow Simulations Over China. *Water Resources Research*, *56*(1), 1–19. https://doi.org/10.1029/2019WR025968

Hock, R. (2003). Temperature index melt modelling in mountain areas. *Journal of Hydrology*, *282*(1–4), 104–115. https://doi.org/10.1016/S0022-1694(03)00257-9

Liang, X., & Guo, J. (2003). Intercomparison of land-surface parameterization schemes: Sensitivity of surface energy and water fluxes to model parameters. *Journal of Hydrology*, *279*(1–4), 182–209. https://doi.org/10.1016/S0022-1694(03)00168-9

Lilhare, R., Pokorny, S., Déry, S. J., Stadnyk, T. A., & Koenig, K. A. (2020). Sensitivity analysis and uncertainty assessment in water budgets simulated by the variable infiltration capacity model for Canadian subarctic watersheds. *Hydrological Processes*, *34*(9), 2057–2075. https://doi.org/10.1002/hyp.13711

Melsen, L. A., & Guse, B. (2021). Climate change impacts model parameter sensitivity-implications for calibration strategy and model diagnostic evaluation. *Hydrology and Earth System Sciences*, *25*(3), 1307–1332. https://doi.org/10.5194/hess-25-1307-2021

Mizukami, N., P. Clark, M., G. Slater, A., D. Brekke, L., M. Elsner, M., R. Arnold, J., & Gangopadhyay, S. (2014). Hydrologic Implications of Different Large-Scale Meteorological Model Forcing Datasets in Mountainous Regions. *Journal of Hydrometeorology*, *15*(1), 474–488. https://doi.org/10.1175/JHM-D-13-036.1

Rakovec, O., Hill, M. C., Clark, M. P., Weerts, A. H., Teuling, A. J., & Uijlenhoet, R. (2014). Distributed evaluation of local sensitivity analysis (DELSA), with application to hydrologic models. *Water Resources Research*, *50*(1), 409–426. https://doi.org/10.1002/2013WR014063

Sobol', I. M., & Kucherenko, S. (2010). A new derivative based importance criterion for groups of variables and its link with the global sensitivity indices. *Computer Physics Communications*, *181*(7), 1212–1217. https://doi.org/10.1016/j.cpc.2010.03.006

Yeste, P., García-Valdecasas Ojeda, M., Gámiz-Fortis, S. R., Castro-Díez, Y., & Esteban-Parra, M. J. (2020). Integrated sensitivity analysis of a macroscale hydrologic model in the north of the Iberian Peninsula. *Journal of Hydrology*, *590*(September 2019), 125230. https://doi.org/10.1016/j.jhydrol.2020.125230

Zegers, G., Mendoza, P. A., Garces, A., & Montserrat, S. (2020). Sensitivity and identifiability of rheological parameters in debris flow modeling. *Natural Hazards and Earth System Sciences*, *20*(7), 1919–1930. https://doi.org/10.5194/nhess-20-1919-2020

---

## Author Comment (AC2)

Replies to reviews

**"Revisiting parameter sensitivities in the Variable Infiltration Capacity model across a hydroclimatic gradient"**

Ulises M. Sepúlveda, Pablo A. Mendoza, Naoki Mizukami and Andrew J. Newman

We thank the Editor and the reviewers for their time in commenting on our paper. We provide responses to each individual point below. For clarity, comments are given in italics, and our responses are given in plain blue text.

**Anonymous Referee #2**

*The authors assessed application results of the VIC model on 101 basins in Chile to test how sensitive model results were to variations in 43 calibration parameters. They explained how some parameters could be adjusted by model users and others were hidden inside the coding of the model. Their results showed that 12 parameters exhibited significant sensitivity in soil, vegetation, and snow input variables. Their work seems to provide insight into the inner workings of the model and to contribute useful guidance for future advancement in the popular VIC model.*

*The paper is long and detailed with an extensive set of graphs and tables that will be of interest to modelers working on the finest details of the VIC model. The writing and presentation are excellent.*

We thank the reviewer for his/her positive feedback.

*The paper would be more useful to hydrologists and modelers who are not part of the mainline VIC users if the authors would include brief information about the model's history and accessibility. Very little information of this nature is included. See Line 142 as an example of how the model is introduced.*

We have included additional information about the model's history and accessibility, following the reviewer's recommendation. Please, see the new updated section 3.1 (L143-L156):

"The Variable Infiltration Capacity (VIC; Liang et al., 1994) model is a semi-distributed, physically-based hydrological model that simulates snow accumulation and melt, evapotranspiration (ET), canopy interception, surface runoff, baseflow, and other hydrological processes at daily or sub-daily time steps. While the model was originally designed as a land surface scheme for coupled simulations within earth system models (Liang et al., 1994), most applications have involved uncoupled simulations for hydrological characterizations and, accordingly, the literature reports many attempts to improve process representations (e.g., Liang et al., 1996, 1999; Cherkauer et al., 2003; Andreadis et al., 2009). VIC is predominantly used in the United States (Addor and Melsen, 2019), with many studies focused on water and energy balances (e.g., Andreadis and Lettenmaier, 2006; Cayan et al., 2010); however, its use has expanded to other geographical domains, including China (e.g., Zhao et al., 2013; Gou et al., 2021), Chile (e.g., Vásquez et al., 2021; Vicuña et al., 2021), Europe (e.g., Lohmann et al., 1998; Roudier et al., 2016) and globally (e.g., Shukla et al., 2013; Yang et al., 2021). Ongoing

community efforts using the VIC model include the NASA Land Information System (LIS; https://lis.gsfc.nasa.gov/, last access: 25 January 2022), NASA's Land Data Assimilation System (LDAS; https://ldas.gsfc.nasa.gov/, last access: 25 January 2022), and the Regional Arctic System Model (RASM; https://www.oc.nps.edu/NAME/RASM.htm, last access: 25 January 2022). This study uses VIC version 4.1.2.g, which can be downloaded from https://github.com/UW-Hydro/VIC/releases, along with other versions".

References:

Addor, N., & Melsen, L. A. (2019). Legacy, Rather Than Adequacy, Drives the Selection of Hydrological Models. *Water Resources Research*, *55*(1), 378–390. https://doi.org/10.1029/2018WR022958

Andreadis, K. M., & Lettenmaier, D. P. (2006). Trends in 20th century drought over the continental United States. *Geophysical Research Letters*, *33*(10), 1–4. https://doi.org/10.1029/2006GL025711

Andreadis, K. M., Storck, P., & Lettenmaier, D. P. (2009). Modeling snow accumulation and ablation processes in forested environments. *Water Resources Research*, *45*, W05429. https://doi.org/10.1029/2008WR007042

Cayan, D. R., Das, T., Pierce, D. W., Barnett, T. P., Tyree, M., & Gershunov, A. (2010). Future dryness in the southwest US and the hydrology of the early 21st century drought. *Proceedings of the National Academy of Sciences of the United States of America*, *107*(50), 21271–6. https://doi.org/10.1073/pnas.0912391107

Cherkauer, K. A., Bowling, L. C., & Lettenmaier, D. P. (2003). Variable infiltration capacity cold land process model updates. *Global and Planetary Change*, *38*(1–2), 151–159. https://doi.org/10.1016/S0921-8181(03)00025-0

Liang, X., Lettenmaier, D. P., Wood, E. F., & Burges, S. J. (1994). A simple hydrologically based model of land surface water and energy fluxes for general circulation models. *Journal of Geophysical Research*, *99*(D7), 14,415.14.428. https://doi.org/10.1029/94jd00483

Liang, X., Wood, E. F., & Lettenmaier, D. P. (1996). Surface soil moisture parameterization of the VIC-2L model: Evaluation and modification. *Global and Planetary Change*, *13*(1–4), 195–206. https://doi.org/10.1016/0921-8181(95)00046-1

Liang, X., Wood, E. F., & Lettenmaier, D. P. (1999). Modeling ground heat flux in land surface parameterization schemes. *Journal of Geophysical Research: Atmospheres*, *104*(D8), 9581–9600. https://doi.org/10.1029/98JD02307

Lohmann, D., Raschke, E., Nijssen, B., & Lettenmaier, D. P. (1998). Hydrologie à l'échelle régionale: II. Application du modèle VIC-2L sur la rivière Weser, Allemagne. *Hydrological Sciences Journal*, *43*(1), 143–158. https://doi.org/10.1080/02626669809492108

Roudier, P., Andersson, J. C. M., Donnelly, C., Feyen, L., Greuell, W., & Ludwig, F. (2016). Projections of future floods and hydrological droughts in Europe under a +2°C global warming. *Climatic Change*, *135*(2), 341–355. https://doi.org/10.1007/s10584-015-1570-4

Sheffield, J., Andreadis, K. M., Wood, E. F., & Lettenmaier, D. P. (2009). Global and continental drought in the second half of the twentieth century: Severity-area-duration analysis and temporal variability of large-scale events. *Journal of Climate*, *22*(8), 1962–1981. https://doi.org/10.1175/2008JCLI2722.1

Shukla, S., Sheffield, J., Wood, E. F., & Lettenmaier, D. P. (2013). On the sources of

global land surface hydrologic predictability. *Hydrology and Earth System Sciences*, *17*(7), 2781–2796. https://doi.org/10.5194/hess-17-2781-2013

Vásquez, N., Cepeda, J., Gómez, T., Mendoza, P. A., Lagos, M., Boisier, J. P., et al. (2021). Catchment-Scale Natural Water Balance in Chile. In *Water Resources of Chile* (pp. 189–208). https://doi.org/10.1007/978-3-030-56901-3_9

Vicuña, S., Vargas, X., Boisier, J. P., Mendoza, P. A., Gómez, T., Vásquez, N., & Cepeda, J. (2021). Impacts of Climate Change on Water Resources in Chile. In *Water Resources of Chile* (Vol. 13, pp. 347–363). https://doi.org/10.1007/978-3-030-56901-3_19

Wang, G. Q., Zhang, J. Y., Jin, J. L., Pagano, T. C., Calow, R., Bao, Z. X., et al. (2012). Assessing water resources in China using PRECIS projections and a VIC model. *Hydrology and Earth System Sciences*, *16*(1), 231–240. https://doi.org/10.5194/hess-16-231-2012

Zhao, Q., Ye, B., Ding, Y., Zhang, S., Yi, S., Wang, J., et al. (2013). Coupling a glacier melt model to the Variable Infiltration Capacity (VIC) model for hydrological modeling in north-western China. *Environmental Earth Sciences*, *68*(1), 87–101. https://doi.org/10.1007/s12665-012-1718-8

---

## Referee Report (RR1)

April 28, 2022
HESS paper review – Sepulveda et al.

The authors have responded to each of the reviewer comments. I believe it is ready for publication.

Since this study looks at a larger number of parameters (soil, vegetation, and snow parameters) whereas all previous VIC parameter sensitivity studies have looked only at soil and maybe also vegetation parameters, it represents an advance and should inform future VIC model parameterization efforts.

I appreciate the analysis of grid cells with snowmelt vs. rainfall driven runoff. It is interesting that soil parameter sensitivity does not change much between the two, as long as the environment is humid.